# Multinucleated Retinal Pigment Epithelial Cells Adapt to Vision and Exhibit Increased DNA Damage Response

**DOI:** 10.3390/cells11091552

**Published:** 2022-05-05

**Authors:** Qin Ke, Lili Gong, Xingfei Zhu, Ruili Qi, Ming Zou, Baoxin Chen, Wei Liu, Shan Huang, Yizhi Liu, David Wan-Cheng Li

**Affiliations:** State Key Laboratory of Ophthalmology, Zhongshan Ophthalmic Center, Sun Yat-Sen University, Guangzhou 510060, China; keqingzzoc@126.com (Q.K.); zhuxingfeigzzoc@163.com (X.Z.); qiruili10@163.com (R.Q.); zouming6991@163.com (M.Z.); chenbaoxin@gzzoc.com (B.C.); lvv961201@163.com (W.L.); huangshan@gzzoc.com (S.H.); liuyizhi@gzzoc.com (Y.L.)

**Keywords:** multinucleation, retinal pigment epithelium, reactive oxygen species, photoreceptor, DNA damage, p53

## Abstract

Multinucleated retinal pigment epithelium (RPE) cells have been reported in humans and other mammals. Rodents have an extremely high percentage of multinucleated cells (more than 80%). Both mouse and human multinucleated RPE cells exhibit specific regional distributions that are potentially correlated with photoreceptor density. However, detailed investigations of multinucleated RPE in different species and their behavior after DNA damage are missing. Here, we compared the composition of multinucleated RPE cells in nocturnal and diurnal animals that possess distinct rod and cone proportions. We further investigated the reactive oxygen species (ROS) production and DNA damage response in mouse mononucleated and multinucleated RPE cells and determined the effect of p53 dosage on the DNA damage response in these cells. Our results revealed an unrealized association between multinucleated RPE cells and nocturnal vision. In addition, we found multinucleated RPE cells exhibited increased ROS production and DNA damage after X-ray irradiation. Furthermore, haploinsufficiency of p53 led to increased DNA damage frequency after irradiation, and mononucleated RPE cells were more sensitive to a change in p53 dosage. In conclusion, this study provides novel information on in vivo PRE topography and the DNA damage response, which may reflect specific requirements for vision adaption and macular function.

## 1. Introduction

The retina pigment epithelium (RPE) is a pigmented cell monolayer between photoreceptors and the choroid of the retina. The RPE plays a key role in normal retina function due to its phagocytosis of photoreceptor outer segments (POS), cycling of retinoids for phototransduction, the constitution of the blood–retina barrier, and the maintenance of the immune-privileged status of the eye [1,2,3]. Therefore, dysfunction of the RPE is closely linked to multiple degenerative diseases of the retina, such as age-related macular degeneration (AMD) [4,5].

Most mammalian cells are mononucleated, while polyploidy is detected in megakaryocytes, hepatocytes, trophoblast giant cells, and cardiomyocytes [6,7]. In humans, multinucleated RPE was reported, where 3–5.3% of human RPE cells are bi-nucleated [8,9]. Interestingly, the percentage of multinucleated RPE is especially high in rodents, compromising more than 80% of total RPE cells, and the amount increases in an age-dependent manner [8,10]. The existence of the multinucleated RPE may be important for the phagocytosis of the RPE since oxidized POS increases RPE multinucleation in vitro [10]. A recent investigation of the human RPE implied that the existence of multinucleated RPE is in accordance with rod and cone photoreceptor density [9]. Therefore, we conducted a comparative study of the multinucleated RPE of nocturnal and diurnal animals with distinct rod and cone percentages in their retinas.

Increased mitochondrial and nuclear DNA damage has been detected in the RPE of degenerated retinas [11,12]. High oxygen tension in the macule and the unique phagocytosis function cause constant oxidative stress in the RPE, which is considered the major insult in DNA damage [12,13]. The efficient repair of a double-strand or single-strand DNA break is critical for preventing the genomic instability that can cause cell death, gene mutation, and cellular senescence. Whether mononucleated and multinucleated RPE cells exhibit altered DNA repair efficiency is largely unknown.

In the present study, we confirmed the existence of multinucleated RPE cells in humans of different ages. The composition of multinucleated RPE cells was further studied in mice from postnatal day 11 (before eye-opening) to 22 months and compared in nocturnal and diurnal animals. Our observation revealed that the multinucleation of RPE cells might be an adaptation to night vision. Finally, we found that multinucleated cells exhibited reduced DNA repair efficiency and were more sensitive to p53 dosage change upon DNA damage exposure in vivo. Haploinsufficiency of p53 leads to delayed DNA damage repair in multinucleated RPE cells compared to mononucleated RPE cells in the same eye.

## 2. Materials and Methods

### 2.1. Animals

C57BL/6J mice were used in this study. Mice were housed in standard cages in a specific pathogen-free facility on a 12-h light/dark cycle with ad libitum access to food and water. Mice were given 1 Gy of X-radiation (Rs2000 160 kV, 25 mA, and 1.22 Gy/min). At the end of the repair time, mice were euthanized, and their eyes were removed. All experimental procedures involving animals were approved by the Animal Use and Care Committee of Zhongshan Ophthalmic Center at the Sun Yat-Sen University, Guangzhou, China. Chickens, pigeons, pigs, and rabbits were bought from the market, zebrafish and rats were bought from Sun Yat-Sen University. For pigeons, chickens, rabbits and pigs, the eyes were harvested within three hours after slaughtering the animals. For mice, rats and zebrafish, the eyes were collected immediately after the death of the animals. The animal ages are: pigeon: 1 year, chicken: 1 year, pig: 10 months, rabbit: 6 months, zebrafish: 1 year, mouse: 2 months, rat: 2 months.

### 2.2. Genotyping

Mouse tail lysis buffer was added to mouse tail, incubating at 65 °C for more than 2 h and heating at 95 °C for 5 min. Briefly, 2 μL of genomic DNA were mixed with 20 μL of the Green Taq Mix (Vazyme, Nanjing, China, #P131-03), For detecting p53 gene, and primers and probes specific for p53-geno (primer-F GTGCCCTGTGCAGTTGTG and primer-R CTCGGGTGGCTCATAAGGTA), p53-neo (primer-F TGAATGAACTGCAGGACGAG and primer-R AATATCACGGGTAGCCAACG). For detecting *Pde6* gene, primer *Pde6brd1* F1: TACCCACCCTTCCTAATTTTTCTCAGC, *Pde6brd1* F2: GTAAACAGCAAGAGGCTTTATTGGGAAC, and *Pde6brd1* R: TGACAATTACTCCTTTTCCCTCAGTCTG were used. For detecting *Crb1* gene, primer *Crb1rd8* F1: GTGAAGACAGCTACAGTTCTGATC, *Crb1rd8* F2: GCCCCTGTTTGCATGGAGGAAACTTGGAAGACAGCTACAGTTCTTCTG, and *Crb1rd8* R: GCCCCATTTGCACACTGATGAC were used.

### 2.3. Fundus Photography and Fluorescein Angiography

Fundus images and fluorescein angiography were performed before X-ray treatment using the Micron IV retinal imaging microscope (Phoenix Research Laboratories, Pleasanton, CA, USA) [14]. After anesthesia with 1% sodium pentobarbital (70 μL/10 g), dilation of the pupils and lubrication of the cornea, the mice were taken for fundus photography first, and then I.P. was injected with 2% fluorescein sodium solution (Al-con laboratories, Fort Worth, TX, USA) (5 μL/g), and fluorescein angiographic images were recorded in 5 min.

### 2.4. Mouse Retina Protein Extraction and Western Blot Analysis

The retinas were dissected in PBS and suspended in 120 μL of RIPA buffer (per retina) containing proteinase inhibitor cocktail (Bimake, Shanghai, China, #B14002), protein phosphatase inhibitor A (Beyotime, Shanghai, China, #P1082) and protein phosphatase inhibitor C (Beyotime, Shanghai, China, #P1092). The total proteins were extracted sonication using an EpiSonic 2000 Sonication System (EPIGENTEK, Farmingdale, NY, USA) (Amplitude: 40%, 10 s on and 10 s off for 7 min in total). For Western blot (WB) analysis, it was performed as described previously with some modifications [14]. For each WB, 30–50 μg of total protein was used. The protein was separated by 12% SDS-PAGE and transferred to the PVDF membrane. The membrane was blocked by 5% milk in TBST for 1 h. After washing with TBST, the membrane was incubated with primary antibodies γH2Ax (Santa Cruz, Dallas, Texas, USA, sc-517348, 1:1000 dilution) and GAPDH (Proteintech, Rosemont, IL, USA, #60004-1-Ig, 1:2000 dilution). The secondary antibody was diluted in TBST (1:3000 dilution). After washing with TBST, enhanced chemiluminescence (ECL) detection was performed by using the Ultra sensitive ECL Chemiluminescence Kit (NCM Biotech, Suzhou, China, #P10300) according to the manufacturer’s specifications. The exposure and development of PVDF membrane were performed using Tanon 5200 (Tanon, Shanghai, China).

### 2.5. Histology, Immunohistochemistry and Immunofluorescence

For immunohistochemistry (IHC) and immunofluorescence (IF), the eyes were fixed in the FAS eye fixation solution (Servicebio, Wuhan, China, #G1109), dehydrated using an increasing ethanol gradient and embedded in paraffin as previously described [14]. Three sections (thickness: 10 μm) through the optic disk of each eye were prepared. The antigen was retrieved by incubation at 95 °C in 10 mM sodium citrate buffer for 15 min, after which the slides were immunoassayed with primary antibodies Rhodopsin (Cell Signaling Technology, Danvers, MA, USA, #27182 1:200 dilution) at 4 °C overnight. The following IHC was conducted according to the manufacturer’s protocol (GTVision TMIII, #GK500705) (Gene Tech, Shanghai, China). After development, the slides were counterstained with hematoxylin and observed under a Tissue-FAXS Q confocal microscope (TissueGnostics, Vienna, Austria). For the immunofluorescence, the slides were immunoassayed with primary antibodies γH2Ax (sc-517348, 1:50 dilution) at 4 °C overnight, followed by a 2-h incubation with the secondary antibody. The cell nucleus was labeled with DAPI (SIGMA, Saint Louis, MO, USA, #D9542). F-actin was labeled with fluorescein isothiocyanate phalloidin (YEASEN, Shanghai, China, # 40735ES75).

### 2.6. Animals’ RPE Flat Mount Immunofluorescence

For mouse RPE IF, the procedure was performed as described previously [15]. The RPE flat mounts were incubated with primary antibodies γH2Ax (sc-517348 1:50 dilution) or 53bp1 (Bethyl, Montgomery, TX, USA, A300-272A-M, 1:200 dilution) overnight at 4 °C, followed by a 2-h incubation with the secondary antibody (Cell signaling # 4412S #8890S) and DAPI (SIGMA #D9542). Images were captured with a Tissue Fax confocal microscope. For chickens, pigeons, zebrafish, pigs, rats, rabbits RPE IF, the cell nucleus was labeled with DAPI, the epithelial cell was labeled with ZO1, or F-actin was labeled with fluorescein isothiocyanate phalloidin (YEASEN #40735ES75). Images were captured with TissueFAXS Q confocal microscope (TissueGnostics, Vienna, Austria). Image J (National Institutes of Health, Bethesda, MD, USA) was used to delineate cell profiles and measure the area of the cell for each age group of mice, more than 50 mono-nucleate RPE cells and multi-nucleate RPE cells were detected. For chickens, pigeons, zebrafish, pigs, rats, or rabbits RPE IF, the cell nucleus was labeled with DAPI, epithelial cell was labeled with ZO1. The slides were captured with a Leica DM4000 B LED (Leica, Wetzlar, Germany) or TissueFAXS Q confocal microscope (TissueGnostics, Vienna, Austria). For images were analyzed by TissueFAXS Viewer (TissueGnostics, Vienna, Austria) and ImageJ (National Institutes of Health, Bethesda, MD, USA).

### 2.7. Comet Assay

The RPE were dissected in PBS and suspended in 1 mL of 0.25% Trypsin for 1 h, 37 °C. After centrifuging and removing Trypsin, the RPE cells were diluted with cold PBS at 1 × 10^5^/mL. Pay attention to avoiding light during the experiment. Using CometAssay^®^ Kit (R&D, Minneapolis, MN, USA, #4250-050-K), the following procedure was conducted according to the manufacturer’s protocol. The cell was labeled with SYBR^®^ GREEN I (biosharp, Hefei, China, #BS358A). Images were captured with a TissueFAXS Q confocal microscope (TissueGnostics, Vienna, Austria). Images analyzed by TriTek Comet Score Freeware 1.6.1.13 (TriTek, Corp. Sumerduck, VA, USA).

### 2.8. Primary Cell Culture

The eyeballs were quickly dipped in 70% ethanol and then rinsed in sterilized PBS. The cornea, lens, iris, and neuron retina were removed and the remaining posterior eyecups in a 1.5 mL EP tube containing 1 mL of pre-warmed Trypsin were added. After incubation at 37 °C for 1 h, we resuspended the RPE cells in the tube by flipping, then gently aspirated the RPE/Trypsin solution to a new tube with a blue tip. Leave the choroid in the original tube. Collected RPE cells by centrifugation. 1500–2000 rpm, RT, 5 min. Washed the RPE pellet with 1 mL pre-warmed complete DMEM 2 times. (1500–2000 rpm, RT, 5 min.) Gently resuspend the washed RPE cells and seed them in the coated 12-well dish with a coverslip. After 5 days, we washed the unattached cells and debris with PBS.

### 2.9. Cell Proliferation Assay

The EdU cell proliferation staining was performed using an EdU kit (BeyoClick™ EdU Cell Proliferation Kit with Alexa Fluor 488, Beyotime Biotechnology, Shanghai, China, C0071S). Briefly, primary mouse RPE cells were seeded in 12-well plates for 5 days. Subsequently, cells were incubated with 10 μM EdU for 4.5 h, fixed with 4% paraformaldehyde for 15 min, and permeated with 0.3% Triton X-100 for another 15 min. The cells were incubated with α-tubulin overnight at 4 °C and followed by a 1h incubation with the secondary antibody in a dark place, later the Click Reaction Mixture for 30 min at room temperature and then incubated with Hoechst 33342 for 30 min. The slides were observed under a 40× oil objective lens with a ZEISS LSM 980 confocal microscope (ZEISS Microscopy, Jena, Germany). Image J was used to count Edu positive or negative cells.

### 2.10. Measurement of Intracellular ROS Levels

The intracellular ROS levels were measured using a Reactive Oxygen Species Assay Kit (Beyotime, Shanghai, China, S0033S). Briefly, the cells were seeded in 12-well plates as described in primary cell culture and exposed to 1 Gy X radiation and continued culture for 4 h. Following the treatment, the cells were incubated with 10 μM DCFH-DA for 30 min at 37 °C and then incubated with Hoechst 33342 for 30 min. The slides were captured with a Leica DM4000 B LED (Leica, Wetzlar, Germany). ImageJ was used to analyze the fluorescence integrity for ROS level.

### 2.11. Mitochondrial Membrane Potential

The mice were divided into three groups, PBS, sodium iodate (SI), and X-ray. Briefly, as for PBS and SI groups, we first intraperitoneally injected mice with PBS or 20 mg/kg SI, and then anesthetized mice with 1% sodium pentobarbital, and then dilated the pupils and lubricated the cornea. Later, 1 μL of 200 μM Mito-Tracker Red CMXRos (Beyotime, Shanghai, China, C1049B) was intravitreally injected and the RPE whole mounts were prepared 1 day-post injection. As for the X-ray group, intravitreal injection of Mito-Tracker Red CMXRos was performed 1 day before exposure to 1 Gy of X-ray irradiation. After dissecting of RPE whole mount, DAPI and FITC were counterstaining as described above. The slides were captured with a Leica DM4000 B LED (Leica, Wetzlar, Germany). Image J was used to analyze mitochondria number and average area.

### 2.12. Statistical Analysis

Results are expressed as mean ± SEM and mean ± SD unless otherwise indicated. GraphPad Prism 9.0software (GraphPad software, Inc., La Jolla, CA, USA) was used for statistical analysis as described in Results. All tests are two-tailed, unpaired *t*-tests unless otherwise indicated. *, *p* < 0.05; **, *p* < 0.01; ***, *p* < 0.0001.

## 3. Results

### 3.1. Distribution of Mononucleated and Multinucleated Cells in Mouse RPE

Firstly, we determined the distribution of mono- and multinucleated RPE cells in mice of different ages. The mice were confirmed by sequencing or PCR to exclude Pde6b^rd1^ or Crb1^rd8^ strains, which are naturally occurring retinal degeneration mouse mutants (Appendix A). The RPE whole mount was obtained from postnatal day 11 (P11), 2-month (2M), and 22-month (22M)-old mice, and image regions were selected according to the distance to the optic nerve head (Figure 1a). Mice have a significantly higher percentage of multinucleated RPE cells compared to humans, and the highest amount was detected around the optic nerve, where 80% of RPE cells were multinucleated in all ages examined (Figure 1b). Furthermore, the number of multinucleated cells decreased in the peripheral regions, which is consistent with previous reports (Figure 1c) [10]. Interestingly, when observed at similar locations, no significant differences in the number of multinucleated cells were found at different ages (Figure 1d), suggesting that region rather than age impacts the existence of the multinucleated RPE in mice. When the RPE cell size was analyzed, multinucleated RPE cells exhibited a two-fold increased area than mononucleated cells in all ages examined (Figure 1e). In addition, significantly increased cell size was observed in old mice (22 M), for both mononucleated and multinucleated RPE cells. Finally, we compared cell proliferation in mononucleated and multinucleated RPE cells by 5-ethynyl-2 deoxyuridine (EdU) analysis. As shown in Figure 1f, mononucleated and multinucleated RPE cells show similar Edu-positive cell percentages when cultured in vitro, suggesting multinucleation does not affect DNA incorporation in RPE cells.

### 3.2. A High Percentage of Multinucleated RPE Correlates with Nocturnal Vision

Although the mouse retina does not have a macula, the central region resembles the human macula in some aspects [15]. Interestingly, the highest frequencies of multinucleated cells in human RPE were found in the macula [9]. Further, multinucleated RPE cells are enriched in macula perifovea, where the highest amount of rods are located but are absent in the macular fovea, which only contains cone photoreceptors [9]. These results prompted us to examine the multinucleated RPE in diurnal and nocturnal vision animals, in which rod and cone photoreceptors show distinct compositions. Generally, nocturnal animals had a higher percentage of rods than diurnal animals [16]. Interestingly, our results show that nocturnal animals (mice, rats, and rabbits) had a significantly increased multinucleated RPE than diurnal animals (chickens, pigeons, pigs and zebrafish) (Figure 2a,b). Even in nocturnal animals, rodents (mice and rats), which have a lower number of cones than rabbits, showed a higher percentage of multinucleated RPE cells than rabbits (Figure 2c). Finally, we found a significant positive correlation between the percentage of multinucleated RPE and rod in six animals with known rod amounts (Figure 2d) [17,18,19,20,21,22,23,24,25,26]. Therefore, we concluded that the percentage of multinucleated RPE and rod photoreceptors is positively correlated in the retina.

### 3.3. Multinucleated RPE Exhibits Increased DNA Damage Compared to Mononucleated RPE

The RPE of mice represents an ideal model system to study DNA damage response in polyploidy cells due to its high percentage of multinucleated cells. Therefore, we induced DNA damage by exposing mice to 1 Gy of X-ray ionizing radiation (IR), as this dosage has been reported to cause DNA double-strand break in mice retinas [27]. We first confirmed that IR led to DNA damage in mice retinas through immunofluorescence (IF) analysis using the DNA damage marker γH2Ax. We found the damaged DNA signal culminated at 1 h post-IR, then dramatically decreased 1 day later and was barely detected 3 days after IR (Figure 3a). This tendency was further confirmed by WB analysis (Figure 3b). Next, we determined RPE DNA damage through γH2Ax staining. Similar to the retina, the RPE displays distinct DNA damage as early as 1 h after irradiation (Figure 3c). The damaged DNA was gradually repaired as γH2Ax-positive cells decreased in number 1 day after IR and further decreased at 3 days post-IR (Figure 3c). Notably, the multinucleated RPE displayed a significantly higher level of γH2Ax-positive cells at all three time points, indicating that the multinucleated RPE may have reduced DNA repair efficiency compared to mononucleated cells (Figure 3c). 53BP1 is a key regulator for DNA damage repair, the 53BP1-decorated nuclear bodies mediate the formation of the DNA damage repair platform. Therefore, we investigated 53BP1 foci in mouse RPE whole mount. However, although multinucleated RPE cells exhibited increased γH2Ax foci, the 53BP1 foci were not significantly altered in multinucleated and mononucleated cells (Figure 3c). Finally, we performed an analysis of DNA double-strand breaks using a neutral comet assay which revealed an increased tail moment 1 h after IR for multinucleated RPE (Figure 3d). Moreover, at 1 and 3 days post-IR, the multinucleated RPE had a 1.32-fold and 1.49-fold greater tail moment than mononucleated cells, respectively (Figure 3e). Together, these results showed that DNA double-strand breaks are repaired less efficiently in multinucleated RPE cells than in mononucleated RPE cells although similar 53BP1 foci formations were observed.

### 3.4. Multinucleated RPE Cells Show Increased ROS Production after IR Exposure

Since increased DNA damage was observed in multinucleated RPE cells, we thus determined reactive oxygen species (ROS), a potent DNA damage inducer for DNA damage in RPE cells. Fluorescent ROS analysis demonstrated that primarily cultured mononucleated and multinucleated RPE cells show similar low ROS levels in normal conditions (Figure 4a,b). After IR exposure, dramatic upregulation of ROS was detected in RPE cells, where multinucleated cells show significantly higher levels of ROS (Figure 4a,b). Since ROS production contributes to mitochondrial damage, we further investigated the mitochondrial morphology in mouse RPE in vivo. As shown in Figure 4c, oxidative stress directly induced by oxidant sodium iodate, or by IR, leads to evident enlarged mitochondria area, possibly due to swelling of mitochondria upon damage insults. However, no significant alterations were found in mononucleated and multinucleated RPE cells (Figure 4d). Taken together, these results indicate that multinucleated RPE cells generated more ROS than mononucleated cells upon DNA damage insult.

### 3.5. p53 Haploinsufficiency Leads to Increased DNA Damage in the RPE

p53 is a key gene controlling the DNA damage response. It is unknown whether or not p53 has a different effect on mononucleated and multinucleated RPE DNA damage. Because homozygous depletion of p53 in C57BL/6J mice led to severe eye abnormalities [28], we used p53 heterozygotes (p53+/−) in our investigation. However, fundus photography revealed that more than 60% of p53+/− mice (14 out of 21 mice) also have an ocular abnormality, including retinal pigment epithelial depigmentation, retina folds, colobomas, and abnormal vasculature (Figure 5a). HE staining further confirmed a retinal fold in those mice (Figure 5b). To exclude the effect of pre-existed ocular abnormalities, we selected p53+/− mice with normal fundus characteristics for IR exposure. IF analysis of RPE flat mounts showed a normal RPE structure in these p53+/− mice (Figure 5c). In control mice, γH2AX signals were barely detected in wild-type (WT) and p53+/− RPE, suggesting that haploinsufficiency of p53 does not spontaneously cause DNA damage (Figure 5c). After IR exposure, p53+/− RPE exhibited increased DNA damage in both mononucleated and multinucleated cells compared to WT RPE, and this higher level of DNA damage was observed at 1 h, 1 day, and 3 days post-IR (Figure 5d,e). These results highlight the requirement of p53 in efficient DNA damage repair in the RPE. Next, we compared DNA damage between mononucleated and multinucleated cells in p53+/− RPE. Similar to WT RPE, p53+/− multinucleated RPE showed a higher level of DNA damage than mononucleated RPE at 1 day and 3 days post-IR. However, when investigated 1 h after IR, mononucleated and multinucleated cells in p53+/− RPE exhibited comparable γH2AX signals (Figure 5f); this contrasts with WT RPE, in which multinucleated cells displayed a higher frequency of DNA damage than mononucleated cells (Figure 4c). These results suggest that mononucleated cells may be more sensitive to p53 reduction than multinucleated cells after DNA damage exposure.

## 4. Discussion

In this study, we determined the composition of multinucleated RPE in several animal species. We revealed region but not age as the determining factor for the multinucleated cell amount in the RPE of mice. The multinucleated RPE is associated with the rods’ percentage, which may be an adaption to nocturnal vision. Moreover, our results demonstrate that a multinucleated RPE has reduced efficiency in DNA damage repair, and p53 dosage change has a stronger impact on mononucleated cells than multinucleated cells after IR-induced DNA damage.

The existence of polyploidy cells might be a consequence of cellular stress or metabolic requirement. For example, polyploidy is important for cardiac muscle function under stressed conditions, and multinucleated mammary epithelial cells are essential for effective lactation [29,30]. Polyploidization may increase tissue metabolic capacity by enhancing transcriptional and translational output [31]. In this regard, the central region of the mouse retina possesses a higher photoreceptor cell density than the peripheral regions, underlining an increased phagocytic and metabolic burden for the central RPE [15]. Our study further correlated the distinct spatial distribution of multinucleated RPE with the rods’ proportion and nocturnal vision adaption. Nocturnal animals have developed several unique ocular structures to maximize light collection; one example is the inverted heterochromatin structure in the rods’ nuclei [22]. Currently, we do not know if a high proportion of multinucleated RPE is required for specific rod photoreceptor metabolism or for dim-light sensing in the dark. Nevertheless, to our knowledge, this is the first evidence linking multinucleated RPE with the rods’ proportion and nocturnal vision. Further studies using neural retina leucine zipper (Nrl) knockout mice, in which rod photoreceptors are converted to cone photoreceptors [32], should directly address this point.

Due to a large number of multinucleated cells, the RPE of mice presents an ideal model to study whether multinucleated and mononucleated cells have different responses to DNA damage. Our results show, for the first time, that the multinucleated RPE cells exhibited increased DNA damage after IR. We speculate that multinucleated RPE cells are more prone to DNA damage in comparison with mononucleated RPE cells, due to the enhanced production of ROS, which is a potent DNA damage inducer. On the other side, we found the key DNA damage repair protein, 53BP1, showed similar foci formation efficiency in mononucleated and multinucleated RPE cells after IR, suggesting the impaired DNA damage repair is not due to the 53BP1 defect. A recent in vitro study showed that multinucleated human RPE1 cells exhibited more γH2AX-marked DNA damage and delayed formation and resolution of 53BP1 foci [33]. In that study, the in vitro cultured cell line was used and multinucleation was induced by disruption of normal cell mitosis [33]. We speculate that different cellular conditions, i.e., naturally occurred versus induced multinucleation, and in vitro versus in vivo environment, may result in different DNA damage responses in diverse multinucleated cells. The exact mechanism responsible for increased DNA damage in multinucleated RPE is still unknown, and systemic analysis of the DNA damage sensors, transducers, and effectors should provide valuable information.

## Figures and Tables

**Figure 1 cells-11-01552-f001:**
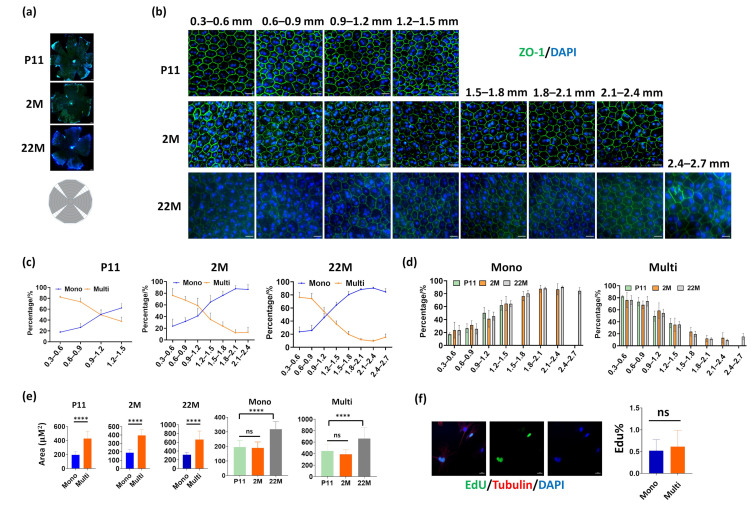
Distribution of mononucleated and multinucleated cells in mouse RPE of different ages. (**a**) IF analysis was performed on whole-mount RPE of postnatal day 11 (mice that had just opened their eyes, P11) 2-month (2 M), and 22-month-old (22 M) mice. The RPE morphology was demonstrated by ZO1 staining, and the nuclei were stained with DAPI. Scale bar: 200 μM for P11 and 500 μM for 2 M and 22 M. The bottom image depicts a schematic graph showing different geographic locations of RPE flat mounts used in image analysis. (**b**) The mononucleated and multinucleated RPE cells are shown in different regions. The length indicates the relative distance from the optic nerve head. Scale bar: 20 μM. (**c**,**d**) Quantification of mononucleated and multinucleated RPE cells at different regions. (**e**) The cellular area of mononucleated and multinucleated RPE cells is indicated. ****: *p* < 0.0001, ns: not significant. For each group, more than 50 cells were quantified. (**f**) EdU staining shows DNA synthesis in primary mouse RPE cell cultures. The cell skeleton structure was labeled by α-tubulin staining and the nuclei were counterstained by DAPI. Right panels: quantification results of EdU-positive RPE cells. For each group, more than 20 cells were quantified. All Data are shown as mean ± SD.

**Figure 2 cells-11-01552-f002:**
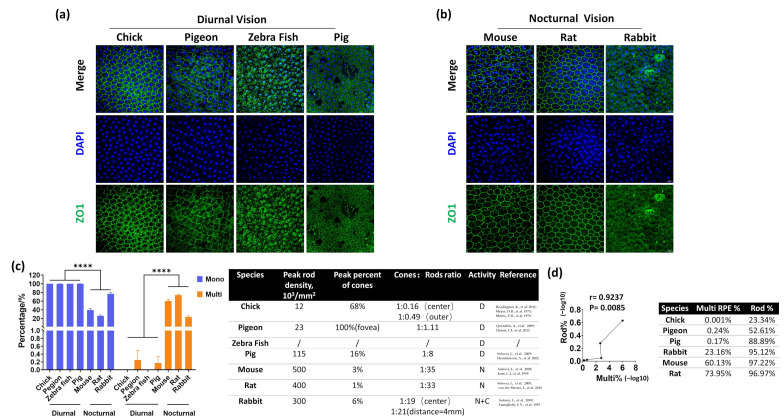
High percentage of multinucleated RPE correlates with nocturnal vision. RPE cells in diurnal vision (**a**) and nocturnal vision (**b**) animals. The PPE morphology was indicated by ZO1 staining, and the nuclei were stained by DAPI. n = 3 for each animal. (**c**) comparison of mononucleated and multinucleated RPE with the key parameters characterizing the adaptation of the retina to nocturnal or diurnal vision. (**d**) Correlation of multinucleated RPE percentage with rods percentage in animals mentioned in (**c**). The data were presented as –log10 and two-tailed Pearson correlation coefficients were used to calculate the r and *p* values.

**Figure 3 cells-11-01552-f003:**
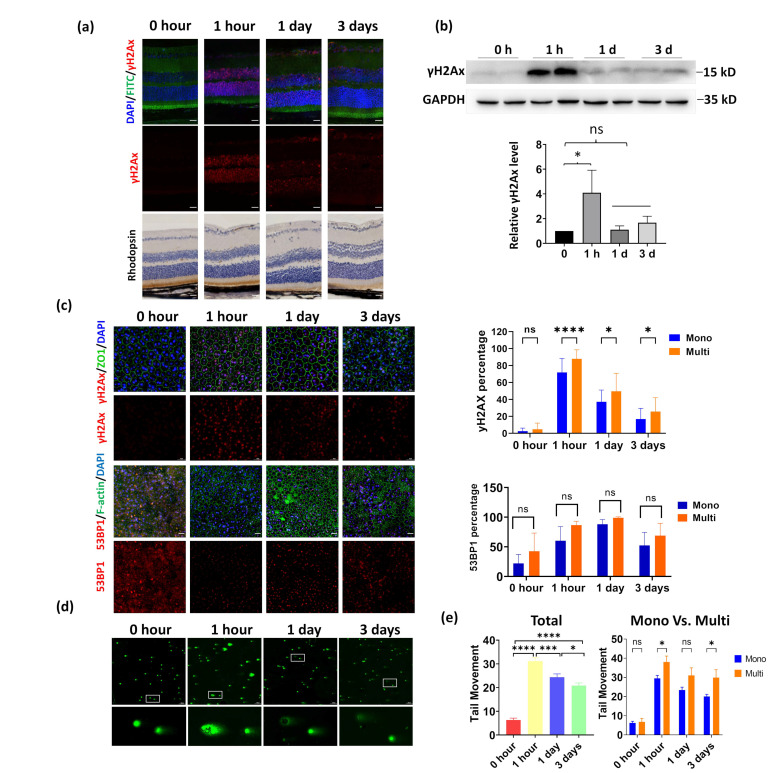
Multinucleated RPE cells exhibit reduced DNA repair efficiency compared to mononucleated RPE. DNA damage was induced by exposing the mice to 1 Gy of X-ray irradiation (IR). The retina and RPE were collected at 1 h, 1 day, and 3 days after IR. (**a**). IF and immunohistochemistry (IHC) analysis of retina cryosections. The DNA damage was indicated by γ-H2AX staining, and the cytoskeleton F-actin was labeled by FITC− phalloidin. The retina structure was further demonstrated by IHC staining of rhodopsin, the photoreceptor marker. Scale bar: 20 μm. (**b**). WB analysis of the retina treated as described above. The quantification results of three independent experiments were shown in the bottom panel. *: *p* < 0.05. ns: not significant (**c**). IF images of γH2AX and 53BP1 staining in the RPE at 1 h, 1 day, and 3 days after irradiation. Scale bar: 20 μm. Right panels: quantitative analysis of the DNA damage comparing mononucleated and multinucleated RPE was completed by counting the γH2AX-positive or 53BP1-positive cells. Forty regions in whole-mount RPE from 4 mice were randomly selected and quantified. * *p* < 0.05, **** *p* < 0.0001 and ns: not significant. (**d**). Comet assay showed DNA damage in RPE cells after IR. Neutral comet assay was performed using digested mouse RPE cells, which were collected at the indicated time point post-IR. The enlarged figure demonstrates a typical multinucleated and mononucleated cell. Scale bar: 100 μm. (**e**). Quantitative analysis of tail movement at the indicated time post-IR. At each time point, more than 80 mononucleated and 15 multinucleated RPE cells were counted, respectively. * *p* < 0.05, *** *p* < 0.001, **** *p* < 0.0001, ns: not significant. All data are shown as mean ± SD.

**Figure 4 cells-11-01552-f004:**
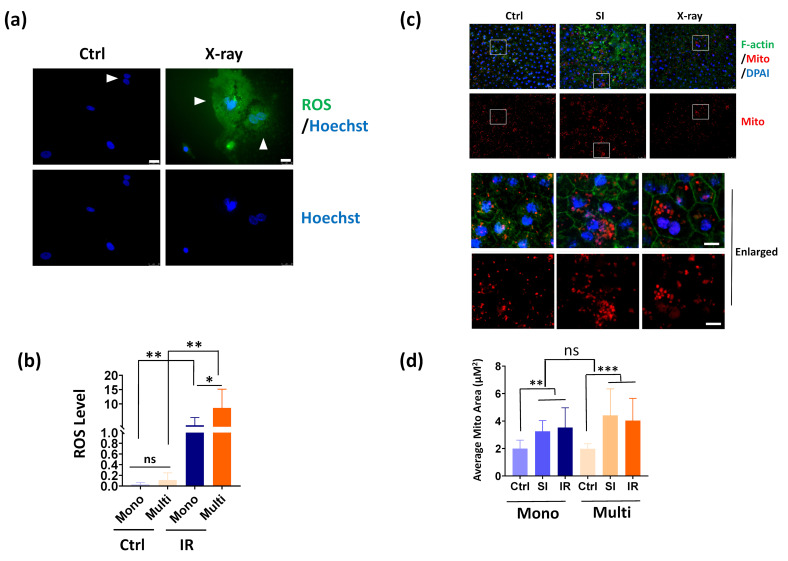
Multinucleated RPE cells generate more ROS than mononucleated RPE cells. Multinucleate mouse RPE cells exhibited increased ROS production than mononucleate cells after IR exposure. The primary mouse RPE cells were untreated (Ctrl) or exposed to 1 Gy of IR. After recovery in growth medium for 4 h, the ROS levels were detected and the nuclei were counterstained by Hoechst. Arrowheads indicate multinucleated RPE cells. (**b**) Quantitative results of ROS as indicated in (**a**). The green fluorescence intensity was quantified by Image J. For each group, about 15 cells were quantified. *: *p* < 0.05, ns: not significant. (**c**) Mitochondria morphology in mouse RPE whole mount with or without sodium iodate (SI) or IR exposure. The F-actin was labeled by FITC-phalloidin and the mitochondria were labeled by Mitotracker red. The nuclei were stained by DAPI. Scale bar: upper panels: 25 μΜ, lower upper panels: 10 μM. (**d**) Quantification results of mitochondria average area in mononucleated and multinucleated RPE cells. n = 15 cells per group. **: *p*< 0.01, ***: *p* < 0.001, ns: not significant. All data are shown as mean ± SD.

**Figure 5 cells-11-01552-f005:**
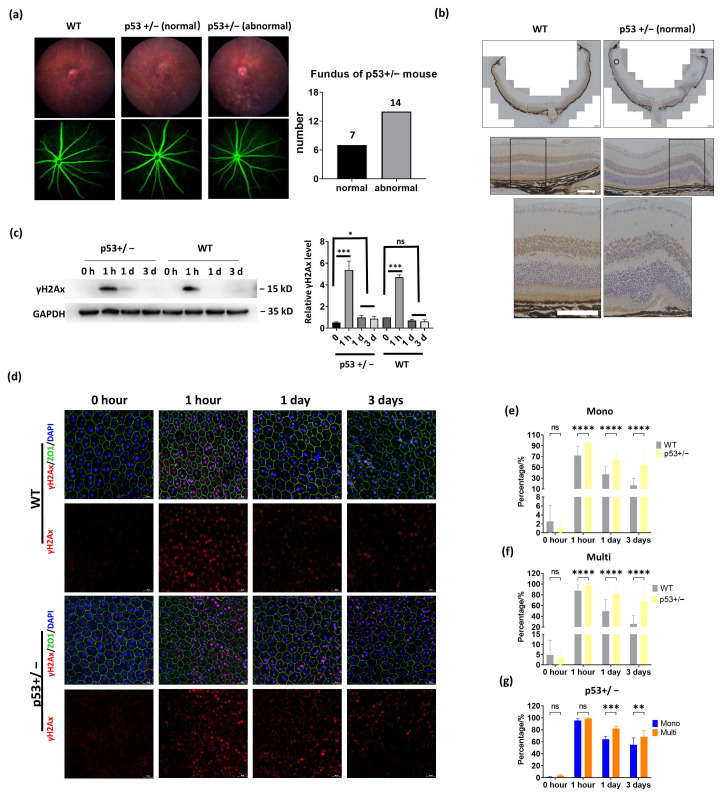
Induction of DNA damage in WT and p53+/− mouse retina. (**a**). Left: fundus photography (upper panels) and fluorescein angiography (lower panels) were performed to analyze WT and p53+/− mice eye morphology. Right: quantification results of the normal and abnormal eye fundus photography from p53+/− mice. (**b**). IHC of rhodopsin shows RPE and retina morphology in WT and p53+/− mice. Scale bar: upper: 200 μΜ, middle and bottom: 100 μΜ. (**c**–**g**). The WT and p53+/− mice were subject to 1 Gy of IR, and retinas or RPE cells were collected at the indicated time point post-IR. (**c**) WB analysis shows relative the indicated protein levels in mouse retinas. Right panels: the relative γH2Ax level was obtained by normalizing with GAPDH. ***: *p* < 0.001, ns: not significant. (**d**). Comparison of DNA damage response in WT and p53+/− RPE. IF analysis of γH2AX staining at the indicated time point post-IR. The cell cytoskeleton F-actin was labeled by FITC-phalloidin staining. Scale bar: 20 μm. (**e**–**g**). Quantitative analysis of the γH2AX-positive RPE cells in WT and p53+/− mice. WT: n = 4, p53+/−: n = 3. Total of 40 regions of WT group and 30 regions of p53+/− group were randomly selected, and the γH2AX-positive cells were counted and quantified. * *p* < 0.05, ** *p* < 0.005, *** *p* < 0.001, **** *p* < 0.0001, and ns: not significant. All data are shown as mean ± SD.

## Data Availability

All data from the study are given in the manuscript.

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
