# Peer review of "Multinucleated Retinal Pigment Epithelial Cells Adapt to Vision and Exhibit Increased DNA Damage Response"

_cells, 2022, doi:10.3390/cells11091552_

Round 1

Reviewer 1 Report

In this article, the authors investigate the occurrence and distribution of multinucleated retinal pigment epithelial (RPE) cells in humans and rodents. They further compare the distribution of multinucleate RPE cells in nocturnal and diurnal animals and reveal a positive correlation between multinucleate RPE and nocturnal vision with rod photoreceptors. This article also provides initial insights into the DNA damage response exhibited by multinucleate RPE cells, which can be useful to further examine degenerative retina. The authors show that multinucleate RPE exhibit increased level of gH2Ax upon damage and compare the effect of p53 dosage upon induction of DNA damage.

General Comments: The manuscript is well written with clear explanation. The data are supportive to the major conclusions.

  • Can the authors comment on whether multinucleate RPE cells have a delayed DNA damage repair response or are prone to more DNA damage in comparison with mononucleate RPE cells?
  • Is this phenomenon of differential DNA damage response in multi vs mononucleate RPE cells observed here, known to occur in other cell types? If so, can the authors discuss that and list some consequences of the same.
  • Can the authors speculate more upon the possible mechanisms that induce differential DNA damage responses in mono vs multinucleate RPE cells?

Specific Comments:

  • Figure 3: What are the ages of the animals used to compare the multinucleate RPE cells? Would the results vary with different ages?

Reviewer 2 Report

The authors orchestrated a very interesting paper about the molecular mechanisms that intervene in vision both in healthy individuals and with eye diseases. I consider the paper interesting and well written and therefore publishable on Cell.

Author Response

We appreciate the reviewer’s positive attitude for our research.

Reviewer 3 Report

In this manuscriptKe and colleagues showed a delayed DNA damage repair in multinucleate RPE cells. The authors claim that depletion of p53 causes increased DNA damage. However, I think it is still necessary for authors to examine the overall function of multinucleate RPE cells to firm up their conclusion.

Authors should study any functional difference (phagocytic and cell proliferation) between mono and binucleate RPE cells. Are the mitochondria of multinucleate RPE cells are more susceptible to cell death? The authors should compare the size of mono and multinucleate RPE cells. As presented, however, the study needs to be strengthened.

Specific comments

Fig1. The authors used a single sample (eye) and counted multiple areas. To get reliable and reproducible data, authors should increase the number of samples (minimum n=3) and analyze the data using appropriate statistical tools. Did the authors study the morphology of RPE cells in these age groups? Is any difference in the proportion of binucleate RPE between peripheral vs central retina?

Fig2. Mice should have been screened for Pde6Brd1 as well as rd8. The age and sex of the mice used should be specified. It’s not clear how many samples (eyes) were analyzed. Authors should analyze data statistically.

Fig. 3. The number of samples used in the study was not clear. Authors should do correlation analysis, and then make statements like “Correlation of mononucleate and multinucleate 192 RPE with the key parameters characterizing adaptation of the retina to nocturnal or diurnal vision.” Scale bar not given.

Fig. 4. The use of additional quantitative assays would greatly bolster this study:

1. ROS production

2. The authors need to provide additional evidence demonstrating that the oxidative stress response directly causes the phenotype and originates in the mitochondria. To me, this is the most novel aspect of the work.

Fig 5. Because homozygous depletion of p53 in C57BL/6J mice already led to severe eye abnormalities], we used p53 heterozygotes (p53+/- ) in our investigation. Therefore, the authors should provide evidence that there is no functional difference between WT and p53+/- ) prior to their studies. In addition, the authors shod show a high mag image of the retina.

Reviewer 4 Report

The manuscript titled “Multinucleation of retinal pigment epithelial cell adapts to vision and exhibits reduced DNA damage repair efficiency” by Ke et.al., investigates the role RPE multinucleation on DNA damage repair. The manuscript requires the following revisions:

  1. The methods section clearly needs to mention how human eyes and eyes from varied species were procured. The authors need to provide appropriate IRB-equivalent approvals for conducting experiments on human tissues.
  2. The methods section needs to clearly mention the post-mortem time for the harvest of tissue.
  3. The authors need to specifically study RPE multinucleation human macular, perifoveal, and far-peripheral regions.
  4. Fig 4B- the western blot should be performed on three biological replicates, and appropriate quantification should be provided.
  5. Figure 4D requires phalloidin and DAPI labeling. The current figure is not publication quality.
  6. Fig 5C requires western blot analysis.
  7. The authors need to procure professional editorial services to improve the overall quality of writing.

Round 2

Reviewer 3 Report

The authors revised the MS  as per the comments